# Comparative Evaluation of Analgesics in a Murine Bile Duct Ligation Model

**DOI:** 10.3390/biomedicines13123034

**Published:** 2025-12-10

**Authors:** Emily Leitner, Tim Schreiber, Hanna Krug, Praveen Vasudevan, Simone Kumstel, Lisa Ernst, René Hany Tolba, Brigitte Vollmar, Dietmar Zechner

**Affiliations:** 1Rudolf-Zenker-Institute of Experimental Surgery, University Medical Centre Rostock, Schillingallee 69a, 18057 Rostock, Germany; 2Institute for Laboratory Animal Science and Experimental Surgery, Faculty of Medicine, RWTH Aachen International University, 52074 Aachen, Germany

**Keywords:** refinement, cholestatic liver disease, burrowing, transmitter implantation, bile duct ligation

## Abstract

**Background:** Reliable analgesia is essential to ensure animal welfare and experimental validity in preclinical disease models. However, evidence on the efficacy and side effects of analgesics remains limited. This study investigated the effects of three commonly used analgesics on animal well-being in a murine model of cholestasis. **Methods:** Thirty male C57BL/6J mice underwent transmitter implantation followed by bile duct ligation (BDL) and received continuous metamizole (3 g/L), tramadol (1 g/L), or carprofen (0.15 g/L) via drinking water before and after surgery. Welfare was evaluated using multiple parameters, including body weight, a distress score, drinking volume, burrowing and nesting behavior, mouse grimace scale (MGS), and telemetric data (heart rate, heart rate variability: SDNN and RMSSD, core body temperature, and locomotion). Additionally, liver and gastrointestinal tissues were analyzed histologically for necrosis and immune cell infiltration. **Results:** Even prior to surgery, analgesic-specific reductions in body weight, drinking behavior, and burrowing and nesting activity were observed. After transmitter implantation, metamizole treatment led to significantly reduced body weight, drinking volume, and locomotion compared to the other two analgesics. Following BDL, all treatment groups exhibited pronounced distress, weight loss, and reduced activity. Tramadol treatment resulted in slightly improved MGS and SDNN values, indicating minor benefits without sustained welfare restoration. In contrast, carprofen treatment was associated with reduced survival and inflammatory alterations in the forestomach. **Conclusions:** None of the tested analgesic regimens fully restored animal welfare after BDL. However, tramadol provided modest advantages, suggesting it may represent the most suitable option among the tested analgesics for the BDL model.

## 1. Introduction

Animal experiments remain an indispensable part of biomedical research, providing essential insights into basic biological processes, disease mechanisms, and potential therapeutic strategies [1,2]. At the same time, increasing public and scientific awareness emphasizes the ethical responsibility associated with the use of laboratory animals [3,4]. To ensure ethical and responsible research, the 3Rs principle (Replace, Reduce, Refine) is applied, urging scientists to replace animal testing whenever possible, reduce the number of animals used to a minimum, and refine experiments to minimize animal suffering as much as possible [5,6].

Reliable refinement requires accurate detection and interpretation of pain and distress in animals. In mice, pain assessment is complicated by their natural tendency to conceal signs of pain and discomfort [7]. Therefore, welfare assessments often rely on multivariable approaches for detecting distress and pain, including behavioral indicators such as burrowing and nesting activity [8,9,10], clinical parameters including body weight and distress score [11,12,13] and physiological readouts, including heart rate (HR), heart rate variability (HRV), body temperature, and locomotion, which can be recorded by telemetry [14]. However, distinguishing between pain and distress remains a major challenge in animals [13,15]. The mouse grimace scale (MGS) is widely used as a non-invasive method to assess pain [16]. Nevertheless, it is known that MGS is not exclusively specific for pain, but can also reflect distress caused by anesthesia [17], fear [18], or discomfort [19]. Similarly to MGS, behavioral and clinical parameters, such as burrowing, nesting, or body weight, cannot reliably distinguish between pain and non-pain-related distress, since changes in these measures may also arise from factors such as anesthesia, fear, or general discomfort [20,21].

Effective pain management is a cornerstone of animal welfare and an essential aspect of experimental refinement [22]. Untreated pain not only compromises animal welfare but also affects experimental outcomes by influencing immune responses, wound healing, tumor progression, as well as endocrine regulation, circadian rhythms, and learning behavior [23,24]. However, selecting appropriate analgesics in preclinical studies remains challenging. Optimal dosages and administration routes are often poorly defined, and limited data exist on how analgesic treatments may interfere with disease progression, thereby complicating their use in translational models [22,25]. To minimize handling stress or stress due to repeated injections [26], oral administration of analgesics such as metamizole or tramadol via the drinking water is frequently used [27,28]. This method allows continuous, non-invasive drug delivery and promotes animal welfare by reducing handling stress. When evaluating analgesics, ethical constraints can also complicate the evaluation of their efficacy in animal models, since the use of untreated controls conflicts with the refinement principle by withholding pain relief during potentially painful procedures [23,29]. Consequently, the focus has shifted to comparative studies of different analgesics under controlled experimental conditions [30].

The challenge of adequate pain management becomes particularly evident in disease models such as cholestasis induced by bile duct ligation (BDL). The BDL model is a well-established and translationally relevant approach for studying human cholestatic liver diseases, including primary sclerosing cholangitis (PSC), a chronic disorder affecting intra- and extrahepatic bile ducts, primary biliary cholangitis (PBC), an autoimmune-mediated destruction of small intrahepatic bile ducts, as well as biliary atresia, a congenital obstruction of extrahepatic bile ducts [31,32]. This model has also proven valuable in evaluating therapies, such as ursodesoxycholic acid, for the treatment of PBC [33]. However, BDL is associated with significant challenges due to high mortality rates [34] and substantial distress [27] in experimental animals. The surgical procedure induces acute pain, while subsequent cholestasis causes systemic inflammation [35], which severely affects the condition of the animals. Consequently, optimizing postoperative pain management in this model is both crucial and challenging.

In many animal models, metamizole (dipyrone), tramadol, and carprofen are used for peri- and postoperative analgesia [36]. Metamizole is a non-opioid analgesic with antipyretic [37] and spasmolytic properties [37] that is frequently used for postoperative pain [38]. Metamizole inhibits peripheral and central prostaglandin synthesis via cyclooxygenase (COX) inhibition [39,40] and modulates the endocannabinoid system through activation of CB1 receptors [41]. Despite its efficacy, metamizole is associated with adverse effects such as hematological alterations, including agranulocytosis [42]. Tramadol is a centrally acting opioid that combines µ-opioid receptor activation with serotonin and norepinephrine reuptake inhibition [43,44]. It provides analgesia for managing moderate to severe pain [45], including postoperative pain. Compared to classical opioids such as morphine, tramadol is associated with a lower incidence of adverse effects [46], although nausea, dizziness, somnolence, and headache may still occur [47]. Carprofen, a nonsteroidal anti-inflammatory drug (NSAID) commonly used in veterinary medicine, primarily inhibits COX-1 and COX-2 enzymes, with a relative preference for COX-2 [48,49].

Despite the widespread use of metamizole, tramadol, and carprofen in rodent research, a direct comparative evaluation of these three analgesics in a cholestatic disease model is currently lacking. Previous comparative studies often focus on postoperative analgesia in acute pain settings, such as laparotomy [14,50], providing important insights into short-term efficacy. However, the BDL model poses additional challenges, as animals experience acute surgical pain and chronic cholestasis-associated distress, including inflammation and hepatocellular injury [51,52]. As chronic inflammation can modify pain perception [53,54] and chronic stress can alter analgesic responsiveness [55,56], analgesic performance may not be directly comparable to short-term postoperative models. The absence of evidence-based comparisons of commonly used analgesics under conditions of combined postoperative and cholestatic pain highlights the gap in refinement research. By combining telemetry with behavioral, clinical, and histological outcomes, the present study addresses this gap and provides an integrated assessment of analgesics’ performance under postoperative and cholestatic conditions.

## 2. Materials and Methods

### 2.1. Ethical Statement

The animal experiments (protocol, research question, key design features, and analysis plan) were approved by the local authority, Landesamt für Lebensmittelsicherheit und Fischerei Mecklenburg-Vorpommern. All experiments were conducted in accordance with the German Animal Protection Law and the European Directive 2010/63/EU [5,57].

### 2.2. Animal Subjects

This study involved 40 male C57BL/6J mice, aged 22 to 32 weeks, which were bred under specific pathogen-free conditions at the University Medical Centre Rostock. Thirty mice were randomly assigned to the analgesia groups (https://stattrek.com), while 10 mice served as controls for histological and blood parameter assessments. One mouse had to be euthanized due to intraoperative complications (intestinal injury) and was excluded from the data analyses, resulting in a final sample size of 29 mice for analgesic comparisons (n(metamizole) = 10, n(tramadol) = 10, n(carprofen) = 9). No control group undergoing surgical procedures without analgesia was included, as such treatment would not comply with ethical standards and the principles of refinement. The sample size was originally calculated when applying for permission to perform these animal experiments (a first calculation was based on published data on body weight as the primary readout parameter with Cohen’s d = 1.9, α = 0.05, β = 0.2, the sample size was then recalculated using first data). The health of all mice bred in the facility was regularly checked in accordance with FELASA guidelines (*Helicobacter* spp., *Rodentibacter pneumotropicus*, and murine norovirus were detected within the last 2 years in the animal facility; unhealthy animals were not used for any experiments). Throughout the experiments, all mice were housed individually in type III cages (Zoonlab GmbH, Castrop-Rauxel, Germany) with a 12 h light-dark cycle (lights on 6 a.m.–6 p.m.), at a temperature of 21 ± 2 °C, and relative humidity of 60 ± 20%. Food (pellets, 10 mm, ssniff-Spezialdiäten GmbH, Soest, Germany) and water were provided ad libitum. Environmental enrichment included nesting material (shredded tissue paper, Verbandmittel GmbH, Frankenberg, Germany), a paper roll (75 × 38 mm, H 0528–151, ssniff-Spezialdiäten GmbH), and a wooden stick (40 × 16 × 10 mm, Abedd, Vienna, Austria).

### 2.3. Assessment of Well-Being

Animal well-being was assessed daily by monitoring body weight, a distress score, and drinking volume using polystyrene pipettes, as described by Bachmanov et al. [58]. The distress score evaluation included various predefined criteria (Appendix A). Mice were euthanized if they exhibited any of the following conditions: persistent cramping, abnormal respiratory sounds, self-mutilation, severe apathy, or a body weight loss of more than 25%. Additionally, euthanasia was performed under isoflurane anesthesia, if the combined distress score exceeded 15 points (with a theoretical maximum of 66 points).

To further assess the well-being of the mice, burrowing behavior was analyzed according to Deacon [8]. A tube (length: 15 cm, diameter: 6.5 cm), filled with 200 g of food pellets, was placed in the cage 3 h before the dark phase. After 2 h, the remaining pellets in the tube were weighed, and the amount of burrowed pellets was calculated. Nesting behavior was evaluated by providing each mouse with a 5 cm square of pressed cotton (ZOONLAB GmbH, Castrop-Rauxel, Germany) 1–2 h before the dark phase. The nesting score was assessed the following morning using a scoring system adapted from Deacon [10]. In addition to 1–5 points from Deacon, we scored 6 points for a perfect nest (with more than 90% of the circumference higher than the animal’s height).

To assess pain and distress, the MGS was applied, as described by Langford et al. [16]. For this assessment, four mice were placed individually in transparent polycarbonate boxes (5 × 5 × 9 cm) inside a light tent with a red background and additional front lighting. After a 3 min acclimatization time, mice were filmed for 10 min using a single-lens reflex camera (Canon EOS 70D, Tokyo, Japan). Six images per mouse were captured at every 1–2 min interval. All images were then blinded and randomized by another person and independently scored by two additional investigators. The mean scores per time point were normalized for each mouse by subtracting the score before analgesia treatment.

Body weight, distress scores, and drinking volume were evaluated daily. For data analyses, body weight data recorded one day after distress scoring were consistently used to assess the impact on weight changes. Burrowing behavior, nesting activity, and MGS scores were assessed multiple times throughout the experiment to monitor their progression and effects over time (Figure 1).

### 2.4. Analgesic Preparation

To assess the impacts of analgesics on distress indicators and side effects, mice were randomly assigned to receive metamizole (Novamin-Ratiopharm^®^, 500 mg/mL, Ratiopharm GmbH, Ulm, Germany), tramadol (Tramal^®^ 100 mg/mL, Grünenthal, Aachen, Germany), or carprofen (Rimadyl^®^, 50 mg/mL, Pfitzer GmbH, Renningen, Germany). The initial administration of analgesia occurred six to 11 days before the telemetry surgery to allow the mice to acclimate. Metamizole (3 g/L, [59,60]), tramadol (1 g/L, [61]), and carprofen (0.15 g/L [61]) were administered daily via drinking water. All mice received one subcutaneous injection of carprofen (5 mg/kg; Rimadyl^®^, Pfizer GmbH, Berlin, Germany) before transmitter implantation and induction of cholestasis in addition to the analgesics administered via drinking water. While this approach likely enhanced overall pain management, it may also have confounded the assessment of the specific effects attributable to the orally supplied analgesics.

### 2.5. Transmitter Implantation and Data Acquisition

To record HR, HRV, body core temperature, and locomotion, ETA-F10 telemetric devices (Data Sciences International, St. Paul, MN, USA) were implanted in mice 22 days before BDL (Figure 1). Anesthesia was induced via inhalation of isoflurane (1–2.5 vol.%, CP-Pharma, Burgdorf, Germany). During the surgical procedure, the mice were placed on a heating pad set to 37 °C, and ophthalmic ointment (Panthenol eye ointment, 50 mg/g; Jenapharm GmbH & Co. KG, Jena, Germany) was applied to protect their eyes. The abdominal area was shaved and disinfected with an iodine solution (Betaisodona^®^, Mundipharma GmbH, Frankfurt am Main, Germany), and a midline laparotomy was performed. The ETA-F10 device was implanted intraperitoneally. The negative electrode was sutured into the *Musculus pectoralis major*, and the positive electrode was attached to the *Musculus externus obliquus abdominis*. The peritoneum was closed with coated 5-0 Vicryl sutures (Johnson & Johnson Medical GmbH, New Brunswick, NJ, USA), and the skin was closed using 5-0 Prolene sutures (Johnson & Johnson Medical GmbH) with a single-knot technique. After the surgery, the mice were placed under a heating lamp for recovery and provided with softened food pellets (ssniff-Spezialdiäten GmbH, Soest, Germany) soaked in water.

After recovery, the mice were placed on receiver plates during the dark phase (6 p.m. to 6 a.m.) on days −22, −21, −15, −2, 0, 1, 4, 7, and 12 to record HR, HRV, body core temperature, and locomotion using the Ponemah software (v5.2; Data Sciences International, St. Paul, MN, USA). Data were collected every minute for each parameter and each mouse. The collected data were analyzed with Ponemah (v6.6; Data Sciences International, St. Paul, MN, USA). HR, standard deviation of all normal-to-normal intervals (SDNN), and the root mean square of successive differences between normal heartbeats (RMSSD) were determined for six 2 min segments during the dark phase. RR intervals with a deviation greater than 3 standard deviations (SD) were excluded from the calculations. The data from these six segments were then averaged.

### 2.6. Induction of Cholestasis

Anesthesia was induced with isoflurane (1–2.5 vol.%, CP-Pharma, Burgdorf, Germany), mice were placed on a heating plate set to 37 °C, and their eyes were protected with ophthalmic ointment. The abdominal area was shaved and disinfected with iodine solution. A midline laparotomy was performed, and the bile duct was carefully dissected from adjacent blood vessels and ligated three times using a 5-0 polyester suture (Polyester-S, Catgut GmbH, Markneukirchen, Germany). The peritoneum was closed with 6-0 polypropylene sutures, and the skin with 5-0 polypropylene sutures (PROLENE^®^, Ethicon Inc., Cincinnati, OH, USA). Following surgery, mice were placed under a heating lamp for recovery and provided with soaked food pellets (ssniff-Spezialdiäten GmbH, Soest, Germany). The entire surgical procedure lasted approximately 25–40 min.

### 2.7. Evaluation of Blood and Tissue

Blood samples were collected via retroorbital puncture under isoflurane anesthesia immediately before euthanasia by cervical dislocation. The samples were centrifuged, and immune cells were analyzed using VetScan HM 5 (Abaxis Europe GmbH, Griesheim, Germany).

The liver and stomach were collected and fixed in 4% paraformaldehyde prepared in phosphate-buffered saline for at least 24 h. To visualize necrosis in the liver, tissue sections of the left liver lobe were stained with hematoxylin and eosin. The necrotic area [%] was quantified in a blinded manner using Image J 1-52a software (National Institutes of Health, Bethesda, MD, USA). To evaluate cellular inflammatory responses, characterized by neutrophilic granulocyte infiltration, 15-naphthol AS-D-chloroacetate esterase (CAE) staining was performed and blindly quantified with ImageJ 1-52a software (National Institutes of Health and Laboratory for Optical and Computational Instrumentation, Madison, WI, USA).

To analyze immune cell infiltration in the stomach and forestomach, hematoxylin-eosin (H&E) staining was performed. For the evaluation, the entire corpus was photographed at 10× magnification, and the photos were taken with 50% overlap. The photos were integrated into an overview picture using Adobe Photoshop CC software v26.5 (Adobe Systems, San Jose, CA, USA) to create a continuous picture for further evaluation. An immune cell infiltration was defined as the accumulation of more than 10 immune cells per field of view and quantified in a blinded manner per nm of the stomach and forestomach using ImageJ software.

### 2.8. Data Analyses

All data were analyzed using GraphPad Prism v8.4.3 (GraphPad Software, San Diego, CA, USA) and are presented as box plots, which show the 5th to 95th percentile range. For this study, no covariates were tested, and statistical significance was determined using appropriate tests based on data distribution (tested with the Shapiro–Wilk test) and the number of independent variables. All statistical tests were two-sided, and differences were considered statistically significant at *p* ≤ 0.05. Behavioral changes and physiological parameters were analyzed using two-way repeated measures ANOVA. Tukey’s multiple comparisons test was applied to compare between analgesic treatment groups, while Dunnett’s test was used for within-group comparisons to baseline. Survival rates were assessed using the log-rank (Mantel–Cox) test. Distress parameters after BDL were analyzed using a mixed-effects model. Tukey’s correction was applied for multiple comparisons between analgesic groups, and Dunnett’s test was used for within-group comparisons to baseline. HR and HRV following transmitter implantation were analyzed using two-way repeated measures ANOVA. Sidak’s multiple comparisons test was applied to assess differences over time (days), and Tukey’s test was used for comparisons between treatment groups. Histopathological alterations between analgesic groups were analyzed using the Kruskal–Wallis test, followed by Dunn’s post hoc test, after normality was excluded using the Shapiro–Wilk test.

## 3. Results

To investigate the effects of different analgesics on animal well-being and postoperative recovery, the study was divided into a pre-phase, an analgesic phase with drug administration via drinking water, a phase after transmitter implantation, and a cholestasis phase (Figure 1). Transmitters were implanted on day −22, followed by a 3-week recovery period before BDL on day 0. Analgesics were administered continuously from the start of the analgesia phase. Welfare parameters were monitored throughout the entire study, while HR, HRV, temperature, and locomotion data were collected from transmitter implantation onward.

### 3.1. Effects of Metamizole, Tramadol, and Carprofen on Welfare Parameters in Healthy Mice

To assess the effects of analgesic treatments independent of surgical or cholestatic intervention, the impact of metamizole (3 g/L), tramadol (1 g/L), and carprofen (0.15 g/L) on welfare parameters was evaluated in healthy C57BL/6J mice (Figure 2). Mice receiving metamizole or tramadol showed a significant transient reduction in body weight compared to baseline values after the initiation of analgesic treatment (A1) (Figure 2A). Metamizole treatment additionally led to a pronounced decrease in drinking volume compared to carprofen-treated mice (Figure 2B). Tramadol administration resulted in reduced burrowing activity relative to baseline, as well as compared to metamizole and carprofen treatment on the initial day of analgesics (Figure 2C). Tramadol also led to a decline in nesting behavior compared to baseline (Figure 2D). In contrast, none of the analgesic treatments significantly affected the MGS or distress score (Figure 2E,F). Thus, even in the absence of any potential painful surgical interventions, analgesics alter body weight, water intake, and certain aspects of mouse behavior.

### 3.2. Metamizole Treatment Is Associated with the Lowest Postoperative Animal Welfare

To continuously monitor HR, HRV, core body temperature, and locomotor activity as physiological indicators of well-being, we implanted a telemetry device in all mice. Since it is known that transmitter implantation transiently affects animal well-being [62], we then evaluated the effects of different analgesics on postoperative recovery over 22 days (telemetry surgery = day −22 relative to BDL).

Implantation surgery caused a temporary reduction in body weight, drinking volume, burrowing activity, and nesting performance, accompanied by a transient increase in distress score and MGS (Figure 3). Metamizole-treated mice exhibited greater body weight loss on day −20 (Figure 3A) and lower drinking volume at several postoperative time points compared to tramadol- or carprofen-treated groups (Figure 3B). Burrowing activity was markedly reduced in tramadol-treated mice relative to those receiving carprofen on day −21 and day −2 (Figure 3D), while carprofen-treated animals displayed enhanced nesting activity compared to tramadol on day −15 and metamizole on day −8 (Figure 3E). Moreover, metamizole-treated mice showed a significantly higher MGS score on day −8 compared to tramadol-treated mice (Figure 3F).

Analyses of analgesic intake via drinking water revealed that most of the metamizole-treated mice did not reach the effective dose levels reported in the literature of 150 mg/kg to 1000 mg/kg [63,64,65,66] during the night following the surgical intervention on day −22 (Figure 3G). For example, 7 from 9 mice did not reach 150 mg/kg/24 h (Appendix A).

In contrast, all 10 of 10 mice exceeded a dose of 20 mg/kg/24 h [67] after transmitter surgery and the recovery phase (Figure 3H, Appendix A).

Carprofen-treated mice also did not achieve the recommended effective dose on day −22 of 20 mg/kg to 25 mg/kg [61,68] (Figure 3I, Appendix A). It should be noted that, in addition to analgesics administered via drinking water, all mice received a single subcutaneous injection of carprofen (5 mg/kg) on day −22 prior to transmitter implantation. Overall, animal welfare was lowest in metamizole-treated animals, whereas carprofen treatment produced minor beneficial effects, such as improved nesting performance.

**Figure 3 biomedicines-13-03034-f003:**
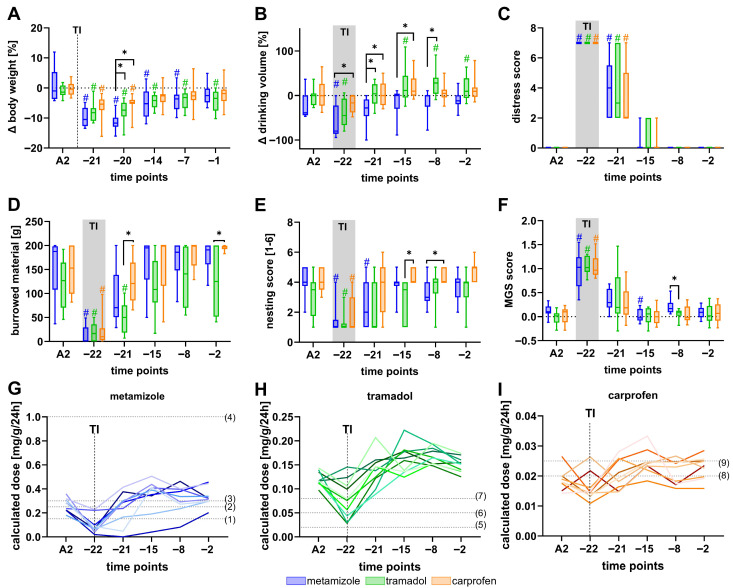
Lowest welfare observed in metamizole-treated mice after transmitter implantation. Transmitter implantation on day −22 reduces body weight (**A**) and drinking volume (**B**), elevates the distress score (**C**), decreases burrowing activity (**D**) as well as nesting behavior (**E**), and increases the MGS score (**F**). Doses of metamizole (**G**), tramadol (**H**), and carprofen (**I**) were calculated for each animal and compared to the published effective doses (published doses = dashed lines, colors indicate individual mice). (1) Stumpf et al. [63], (2) Taylor et al. [65], (3) Schlosburg et al. [64], (4) Boros et al. [66], (5) Sacerdote et al. [67], (6) Mouedden et al. [69], (7) Wolfe et al. [70], (8) Matsumiya et al. [68], and (9) GV-Solas Committee [61]. Data after transmitter implantation (TI), but 2 to 22 days before BDL, were compared to the last measurement before transmitter implantation (A2). Statistical analyses included two-way repeated measures ANOVA with Tukey’s correction for multiple comparisons between analgesic groups (* *p* ≤ 0.05), and Dunnett’s post hoc test for within-group comparisons relative to baseline (# *p* ≤ 0.05). n (metamizole) = 10), n (tramadol) = 10, n (carprofen) = 9.

To further investigate physiological recovery, HR, HRV (SDNN, RMSSD), core body temperature, and locomotion were recorded by telemetry (Figure 4). Notably, HR was increased significantly immediately after surgery (d −22) compared to day −2, a time point where the animals had recovered for over 20 days (Figure 4A). Additionally, both the SDNN (Figure 4B) and RMSSD (Figure 4C) were significantly reduced on day −22 compared to day −2. Interestingly, mice treated with carprofen exhibited significantly lower SDNN and RMSSD values than those receiving metamizole on day −2. Core body temperature remained largely stable throughout the experiments, with no significant differences between the groups (Figure 4D). Locomotor activity was decreased significantly in all groups on day −22 compared with day −2. Furthermore, metamizole-treated mice exhibited significantly lower locomotion than the tramadol or carprofen groups on day −15, and lower than the tramadol group on day −8 (Figure 4E).

### 3.3. Welfare Impairment Persists Despite Analgesic Treatment After BDL

To assess postoperative outcome and overall survival following BDL, Kaplan–Meier survival curves were generated for each treatment group (Figure 5A). Mice treated with metamizole or tramadol showed the highest survival probability during the 14-day observation period (70%). In contrast, carprofen-treated mice demonstrated the lowest survival rate, with a decline in survival within the first week, and a final survival probability of approximately 33%. The odds ratio for survival in metamizole- and tramadol-treated animals compared to carprofen was 4.7 (Figure 5B).

Following BDL, all groups exhibited a pronounced and sustained decline in animal welfare compared with baseline values (day −1 for body weight, day −2 for all other parameters; Figure 6). Body weight significantly decreased in all groups and continued to decline over the observation period without signs of recovery (Figure 6A). Drinking volume dropped markedly on the day of surgery (day 0) in metamizole and tramadol groups compared to baseline and gradually increased thereafter (Figure 6B). The distress score remained significantly elevated after BDL and throughout the study in all groups (Figure 6C). Burrowing activity (Figure 6D) and nesting behavior (Figure 6E) were significantly reduced post-BDL and showed no substantial improvement over time. Similarly, the MGS increased significantly in all groups following BDL compared to baseline and remained elevated throughout the 14-day period (Figure 6F). Notably, tramadol-treated animals exhibited a significantly lower MGS score on day 7 after BDL compared to the carprofen group.

To evaluate potential differences in drug exposure, the effective daily intake of each analgesic was calculated based on individual drinking volumes and compared to effective dosages in the literature (Figure 6G–I). It should be noted that, in addition to analgesics administered via drinking water, all mice received a single subcutaneous injection of carprofen (5 mg/kg) prior to BDL. Most metamizole-treated mice failed to reach the recommended effective dose range reported in the literature of 150–1000 mg/kg [63,64,65,66] (Figure 6G, Appendix A). Especially on the surgery day (day 0), most mice did not reach the recommended dose levels (Appendix A). In contrast, the majority of tramadol-treated mice reached or exceeded the recommended dose range of 20–80 mg/kg [67,69,70], just two mice were below 20 mg/kg on day 0 (Figure 6H, Appendix A), whereas most carprofen-treated mice did not drink enough to reach the effective dose ranges of 20–25 mg/kg [61,68] (Figure 6I, Appendix A).

Additionally, BDL caused significant alterations in physiological parameters across all analgesic groups as measured by the implanted transmitter (Figure 7). HR increased on the day of BDL surgery in all groups, peaking on day 1 and subsequently declining by day 4 (Figure 7A). By day 12, tramadol-treated mice exhibited significantly higher HR values than carprofen-treated animals. SDNN decreased significantly following BDL and remained suppressed throughout the 14-day observation period (Figure 7B). However, SDNN was significantly higher in tramadol-treated mice than in metamizole- and carprofen-treated groups on day 4 and compared with carprofen on day 7. RMSSD also decreased significantly after BDL and remained low in all groups without intergroup differences (Figure 7C). Core body temperature was significantly reduced in all groups compared with baseline values and did not normalize during the 14 days (Figure 7D). Similarly, locomotor activity declined sharply after surgery and remained consistently low across all treatment groups (Figure 7E).

### 3.4. Histopathological Evaluation of the Stomach and Liver After 14 Days of Cholestasis

To further evaluate potential side effects of analgesics, histological analyses of the stomach and liver were performed (Figure 8). Immune cell infiltration in the forestomach (non-glandular) was most pronounced in carprofen-treated mice, showing significantly higher levels compared with control and metamizole-treated groups (Figure 8A,B). In the glandular stomach, mild inflammatory infiltrates were present in all groups, but quantitative analyses revealed no significant differences (Figure 8C,D).

Liver histopathology confirmed extensive necrotic areas in all cholestatic animals treated with metamizole, tramadol, or carprofen compared with healthy controls (Figure 8E,F). However, no significant differences were observed between the analgesic treatments. To further assess hepatic inflammation, CAE staining was used to quantify neutrophilic granulocyte infiltration. CAE-positive areas were significantly increased in all treatment groups relative to controls, but without intergroup differences (Figure 8G,H).

## 4. Discussion

### 4.1. Sensitivity of Welfare Indicators

This study demonstrated that all methods employed were sufficiently sensitive to detect impaired welfare in mice following surgical interventions (Figure 3, Figure 4, Figure 6 and Figure 7). For instance, we observed reduced burrowing and nesting behavior after transmitter implantation and induction of cholestasis, indicating compromised well-being. This finding aligns with previous reports showing that these behaviors can be negatively affected by pain [71] or stress [72]. However, while these changes are sensitive indicators of reduced welfare, they do not allow a clear differentiation between pain and non-pain-related distress.

Similarly, physiological parameters such as HR and HRV provide a measure of autonomic nervous system activity and offer insights into an animal’s physiological response to distress [73]. After both surgical interventions, we observed an increase in HR and a reduction in HRV parameters (SDNN, RMSSD). The increased HR and reduced HRV are associated with sympathetic dominance and reduced vagal tone, which reflects the response to postoperative pain and stress [14,74]. Similarly, clinical indicators such as piloerection [75], changes in water intake [61,75], and body weight [11] have also been reported as reliable but non-specific signs of impaired health or welfare. For instance, body weight loss may also result from general distress or fear rather than pain alone [21].

The MGS, a frequently used tool for facial pain assessment [16], was significantly elevated following both transmitter implantation and BDL. However, it is well described that pain-free conditions, such as distress caused by, for example, anesthesia [17], fear [18], and discomfort [19], can also increase the MGS score. Thus, an increased MGS may indicate either pain or non-pain-related distress.

As all these indicators are sensitive to overall welfare impairment but lack specificity for pain, it is not possible to determine precise pain levels during the experiments. Consequently, we can only infer a general reduction in welfare following transmitter implantation or BDL. This limitation is shared by other studies relying on comparable welfare assessment approaches [27,62], highlighting the need for more specific tools to distinguish pain from distress in laboratory animals. Accordingly, differences in these indicators observed after administration of a specific analgesic are more appropriately interpreted as improved animal welfare rather than a specific reduction in pain.

### 4.2. Effects of Analgesics on Healthy Mice

Two of the analgesics evaluated in this study affected multiple welfare-related parameters, even in the absence of surgical or disease-induced pain (Figure 2).

Metamizole-treated mice showed a significant reduction in body weight and drinking volume compared to baseline values. This observation is consistent with previous reports in C57BL/6J mice, where oral metamizole led to decreased fluid intake and weight loss, for example, in models of colitis [28] and pancreatitis [76]. Such findings raise concerns about the suitability of metamizole for oral administration, as reduced drinking behavior can reduce welfare and limit analgesic delivery.

Tramadol administration resulted in decreased burrowing and nesting behavior. Similar results were observed by Gould et al. in sham-operated rats, where high doses of tramadol reduced burrowing activity, likely due to the sedative properties of tramadol [77]. Additional studies in rodents have shown delayed withdrawal reflexes at 25 mg/kg tramadol and reduced locomotion, confirming that sedation might be a frequent side effect [78,79]. These drug-related alterations pose a significant challenge in evaluating opioid-induced analgesia in animal studies, as they can reduce animal activity, complicating the interpretation of results.

In contrast, carprofen administration did not significantly affect body weight, burrowing activity, nesting behavior, MGS, or distress score, suggesting that it does not interfere with welfare indicators in healthy animals.

Overall, these findings indicate that analgesics can significantly influence welfare measures, emphasizing the need to evaluate their pharmacological effects in healthy mice to ensure accurate interpretation of distress and recovery in research settings.

### 4.3. Recovery After Transmitter Implantation and Analgesics-Specific Effects

Telemetry implantation is a moderate surgical intervention that transiently affects welfare in rodents. In our study, it led to short-term reductions in body weight, drinking volume, burrowing and nesting activity, as well as a temporary increase in distress score and MGS (Figure 3). Comparable results were reported by Kumstel et al., who observed transient reductions in body weight, burrowing and nesting behavior, and an increase in the distress score following transmitter implantation [62]. Similarly, Helwig et al. demonstrated that radiotelemetry implantation temporarily reduced water intake and body weight and suppressed wheel running activity, emphasizing the short-term impact of surgery on welfare parameters [80].

Although transmitter implantation was followed by a 22-day recovery period in our study, minor residual effects persisted. While the distress score, burrowing, and nesting had returned to baseline in all groups, mice in the tramadol group still showed reduced body weight and increased water intake even at the end of the recovery period compared with their baseline values. These deviations suggest that subtle long-term effects of transmitter implantation or tramadol exposure cannot be fully ruled out and should be considered when interpreting distress caused by BDL.

Telemetry recordings further revealed that transmitter implantation induced a pronounced autonomic stress response [81], as indicated by increased HR and reduced HRV (SDNN, RMSSD) on day −22 compared to post-recovery baseline at day −2 (Figure 4A–C). Notably, HRV metrics remained lower on day −2 in carprofen-treated animals than in those receiving metamizole. Given the extended recovery period, these differences are more plausibly explained by drug-specific autonomic modulation than by persisting postoperative pain. COX-2 inhibitors such as carprofen reduce the synthesis of prostaglandins (e.g., PGE_2,_ PGI_2_) that normally support vascular tone [82], renal perfusion [83], and blood pressure homeostasis [84]. Their inhibition due to COX-2 inhibition can lead to elevated arterial pressure [85] via increased vasoconstriction and sodium and water retention [86,87]. These hemodynamic alterations can lead to a compensatory autonomic response that reduces HRV [88]. In contrast, metamizole has been reported to exert vasodilatatory effects [89], potentially promoting higher HRV. These pharmacodynamic differences might explain the pronounced reduction in HRV in carprofen-treated mice. Importantly, such interactions underscore that HRV cannot be interpreted solely as a distress marker when comparing drugs with different cardiovascular profiles.

Among the tested analgesics, metamizole-treated animals showed a greater body weight loss and decreased drinking volumes compared to the other analgesics. Many animals did not reach published effective dose ranges (150–1000 mg/kg) [63,64,66], especially on days of surgery (Figure 3G, Appendix A). Furthermore, metamizole-treated mice showed significantly reduced locomotion at certain time points during the recovery phase compared with mice treated with tramadol or carprofen (Figure 4E).

Tramadol administration resulted in reduced burrowing and nesting activity compared to carprofen-treated mice (Figure 3D,E), likely reflecting its sedative properties as discussed in Section 4.2. Thus, sedation may mask the recovery, complicating a straightforward interpretation of these results.

In contrast, carprofen treatment did not lead to any significant reduction in animal welfare compared with mice provided with metamizole- or tramadol-supplemented drinking water. These mice also seemed to have a faster recovery after transmitter implantation, showing earlier normalization of body weight and behavioral activity compared to metamizole- and tramadol-treated groups. Similar findings, supporting carprofen as a more effective postsurgical option compared to tramadol, were reported in the literature. For instance, Zegre Cannon et al. and Delgado et al. also found carprofen to be more effective than tramadol in managing postoperative pain [90,91]. Only a few comparisons were published between carprofen and metamizole. For instance, Talbot et al. found that postsurgical distress in mice was comparable between those treated with carprofen or metamizole [92].

Taken together, our findings indicate that metamizole is less suitable for postoperative analgesia following transmitter implantation. In addition, none of the tested analgesics fully prevented transient welfare impairment.

### 4.4. Analgesic Treatment Is Insufficient to Completely Prevent Welfare Impairment in the BDL Model

The BDL model, as a highly invasive procedure followed by progressive cholestasis, is associated with significant distress [27]. In our study, all mice showed significant and persistent impairments in welfare after BDL, reflected by sustained weight loss, elevated distress scores and MGS, and markedly reduced burrowing and nesting behavior (Figure 6). Physiological parameters recorded by telemetry, such as increased HR and reduced HRV, further confirmed postoperative stress-induced sympathetic activation [14] (Figure 7). These findings demonstrate that none of the tested analgesic regimens provided sufficient relief to restore welfare completely.

Tramadol-treated mice showed better survival rates than those treated with carprofen. In addition, mice under tramadol had slightly lower MGS scores (Figure 6F) and transiently higher SDNN (Figure 7B). Although these results highlight modest, parameter-specific advantages of tramadol, the overall dataset does not provide strong evidence that it is substantially more beneficial than the other analgesics tested.

Mice treated with metamizole also exhibited higher survival rates compared to those receiving carprofen. However, some mice failed to reach the effective dose range (150 to 1000 mg/kg based on published nociceptive testing data [63,64,66]) more than 24 h after BDL, at a time point when the single s.c. Injection of carprofen was possibly no longer effective (Appendix A). This may have been caused by the observed postoperative reduction in water consumption, illustrating a major methodological limitation of providing analgesics via the drinking water.

In contrast, carprofen was associated with the lowest survival rate (33%) and pronounced inflammatory changes in the forestomach (Figure 8A,B). Although NSAID-related gastrointestinal toxicity is well documented [93,94], carprofen may be particularly susceptible to cholestasis-related exacerbation of toxicity due to its predominantly biliary elimination [95]. Following BDL, cholestasis markedly reduces bile flow [96], which likely impairs the clearance of carprofen and its metabolites. Elevated exposure in combination with cholestasis-associated hepatic injury may further disrupt cholestasis-induced mucosal integrity [97] and enterohepatic circulation [98]. These changes likely amplify NSAID-induced gastrointestinal toxicity and indicate that carprofen may be unsuitable for use in cholestatic disease models.

Overall, these results indicate that although tramadol provided the most favorable welfare outcomes, it did not completely mitigate cholestasis-induced distress.

### 4.5. Methodological Considerations and Limitations

This study provides a comprehensive analysis of animal welfare following transmitter implantation and BDL by combining behavioral, clinical, and physiological parameters. However, several methodological aspects must be acknowledged when interpreting the results.

First, the absence of an untreated (analgesic-free) control group limits our ability to draw conclusions about the absolute analgesic efficacy of each drug. Including such a control would require inducing pain in animals without providing analgesic treatment. This is an ethically problematic approach that, in most European countries, must be explicitly approved by the animal welfare authorities before experimentation. Since permission to include such a control was not granted, we could not determine the absolute extent of pain reduction achieved by these analgesics. Consequently, the aim of this study was to perform a relative comparison of three commonly used analgesics rather than to assess their absolute efficacy.

Second, oral administration via drinking water, while minimally invasive and refinement-oriented, introduces considerable variability in dosing (Appendix A). Following transmitter implantation and BDL, drinking volume markedly declined, leading to inter-individual differences in analgesic uptake. Consequently, animals may have received subtherapeutic or inconsistent doses, particularly in the metamizole and carprofen groups (Figure 3G–I, Figure 6G–I, Appendix A). Such variability is highly relevant for interpreting the effects of analgesics on welfare parameters, as differences may reflect variable drug exposure rather than analgesic differences. Animals with low water intake may appear to exhibit poor analgesic response to the drug, whereas those with high intakes may display drug-related side effects. This limitation must be taken into account when evaluating group differences.

Please note that we administered a single subcutaneous injection of carprofen (5 mg/kg) before transmitter implantation and BDL in addition to the analgesics provided in the drinking water. Our data (Appendix A) support the necessity of this additional perioperative treatment, and future studies should assess whether higher or repeated doses could improve analgesic coverage. The extra injection introduces a pharmacological confound that must be considered when interpreting early postoperative analgesic differences. As carprofen has a reported subcutaneous elimination half-life of approximately 8–10 h [99,100] in mice, it likely overlapped with the onset of oral analgesics, potentially leading to transient additive or synergistic effects. Previous studies indicate that opioids and NSAIDs can exhibit synergetic analgesic interactions [101,102]. Consequently, early postoperative effects observed in the tramadol group cannot be attributed solely to tramadol. Moreover, combined subcutaneous and oral administration of carprofen may lead to carprofen accumulation post-BDL, potentially contributing to adverse effects, given the partial biliary elimination [95]. These interactions should be considered when interpreting the acute postoperative phase.

Another limitation of our study is its exclusive focus on male C57BL/6J mice, as analgesic efficacy and pain sensitivity are known to vary substantially across sexes and mouse strains [103]. Numerous comparative studies have demonstrated differences in nociceptive responses and analgesic sensitivity among inbred strains, including C57BL/6J, BALB/c, and others [104]. Sex-related variation has also been widely reported, partly driven by differences in pharmacokinetics due to distinct body fat distribution between males and females [105], immune response, activity level [103,106], or gonadal hormones [107]. These factors can alter pain sensitivity and analgesic responses, with some drugs like morphine showing pronounced sex-dependent effects [108]. Therefore, our findings cannot be generalized to female mice or other genetic backgrounds, highlighting the need to include both sexes and additional strains when evaluating refinement strategies.

Overall, this study exemplifies how adhering to high ethical standards can impose limitations that may reduce the scientific knowledge gained from research projects, while simultaneously giving rise to new ethical dilemmas.

## 5. Conclusions

This study demonstrates that optimizing analgesia in animal experiments remains a considerable challenge. Depending on the experimental model, substantial differences between analgesics were observed regarding welfare outcomes, pharmacological side effects, and recovery processes. Carprofen appeared to facilitate postoperative recovery following transmitter implantation but was associated with increased mortality and forestomach inflammation under cholestatic conditions. Tramadol yielded slightly better outcomes in the BDL model, reflected by lower MGS and higher SDNN values. However, its overall impact also remained insufficient to ensure optimal welfare. Metamizole proved least beneficial throughout the study, as evidenced by pronounced body weight loss and reduced water intake. These findings underscore the importance of continued research to refine analgesic strategies by systematically evaluating animal welfare, dosage, and administration regimens across different experimental models.

## Figures and Tables

**Figure 1 biomedicines-13-03034-f001:**
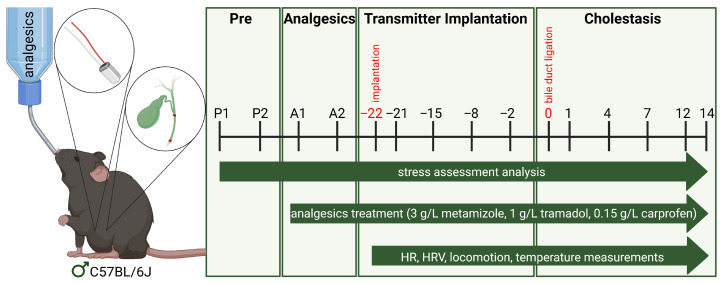
Experimental protocol. The experiment consisted of four phases: During the pre-phase, baseline values were collected at P1 (39 to 34 days before BDL) and P2 (36 to 31 days before BDL). In the analgesic phase, mice received metamizole (3 g/L), tramadol (1 g/L), or carprofen (0.15 g/L) in their drinking water throughout the experiment (initial day of analgesia: A1 = 33 to 28 days before BDL; last day before transmitter implantation: A2 = 23 days before BDL). The telemetry phase started on day −22, allowing continuous monitoring of heart rate (HR), heart rate variability (HRV), body temperature, and locomotion. Cholestasis was then induced by BDL on day 0. Welfare was assessed by burrowing and nesting behavior, body weight changes, a distress score, mouse grimace scale (MGS), and drinking volume. Blood and tissue samples were collected 14 days after BDL (Created in BioRender.com. Leitner, E. (2025). https://app.biorender.com/illustrations/67ced885b963784168b43b94?slideId=64c87ded-0dce-42eb-90e2-70cb0dd5e0a4).

**Figure 2 biomedicines-13-03034-f002:**
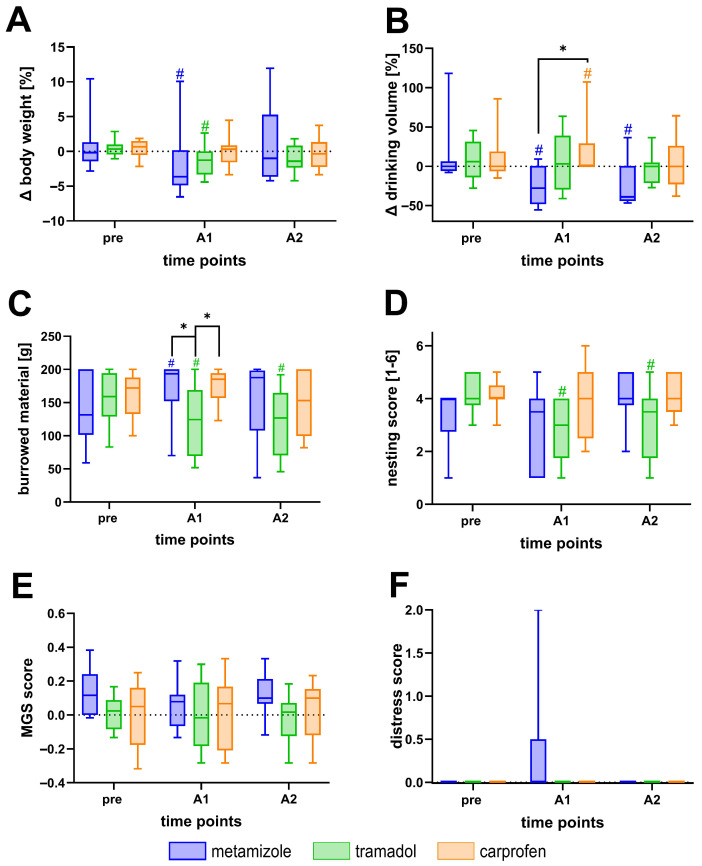
Effects of analgesics on healthy C57BL/6J mice. Body weight (**A**), drinking volume (**B**), burrowing activity (**C**), nesting behavior (**D**), MGS (**E**), and distress score (**F**) were monitored during the analgesic phase. Mice received metamizole (3 g/L), tramadol (1 g/L), or carprofen (0.15 g/L) in drinking water. Data were collected at baseline (pre), after initiation of analgesic treatment (A1 = 33 to 28 days before BDL), and before transmitter implantation (A2 = 23 days before BDL). Statistical analyses were performed using two-way repeated-measures ANOVA followed by Tukey’s post hoc test for multiple comparisons between treatment groups (* *p* ≤ 0.05) and Dunnett’s post hoc test for comparisons with baseline (# *p* ≤ 0.05). n (metamizole) = 10, n (tramadol) = 10, n (carprofen) = 9.

**Figure 4 biomedicines-13-03034-f004:**
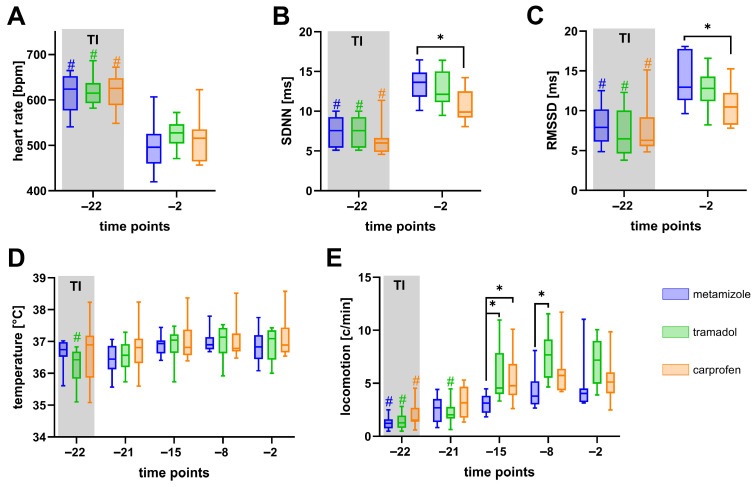
Effects of analgesics on physiological parameters after transmitter implantation. HR (**A**) significantly increased, and SDNN (**B**) and RMSSD (**C**) decreased significantly after implantation (day −22) compared to day −2. Core body temperature (**D**) and locomotion (**E**) were also monitored. Data were analyzed using two-way repeated measures ANOVA with Sidak’s correction for HR, SDNN values, and RMSSD. Dunnett’s post hoc correction was used for within-group comparisons relative to baseline (# *p* ≤ 0.05). Tukey’s test was used for inter-group comparisons (* *p* ≤ 0.05). n (metamizole) = 10, n (tramadol) = 10, n (carprofen) = 9.

**Figure 5 biomedicines-13-03034-f005:**
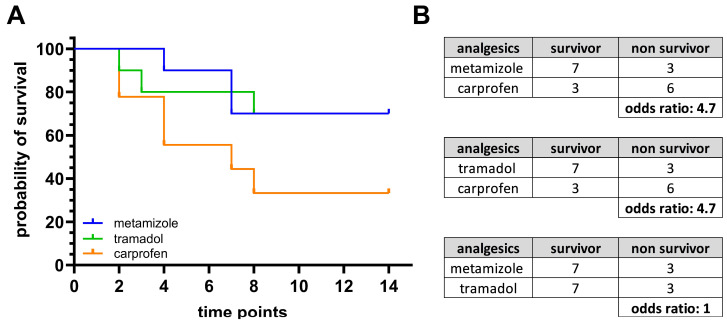
Survival of C57BL/6J mice following BDL under different analgesic treatments. Probability of survival (**A**) of mice treated with metamizole, tramadol, or carprofen after BDL. Odds ratios (**B**) comparing survival between the treatment groups: metamizole vs. carprofen, tramadol vs. carprofen, and metamizole vs. tramadol. Statistical analysis was performed using the log-rank Mantel–Cox test. N (metamizole) = 10, n (tramadol) = 10, n (carprofen) = 9.

**Figure 6 biomedicines-13-03034-f006:**
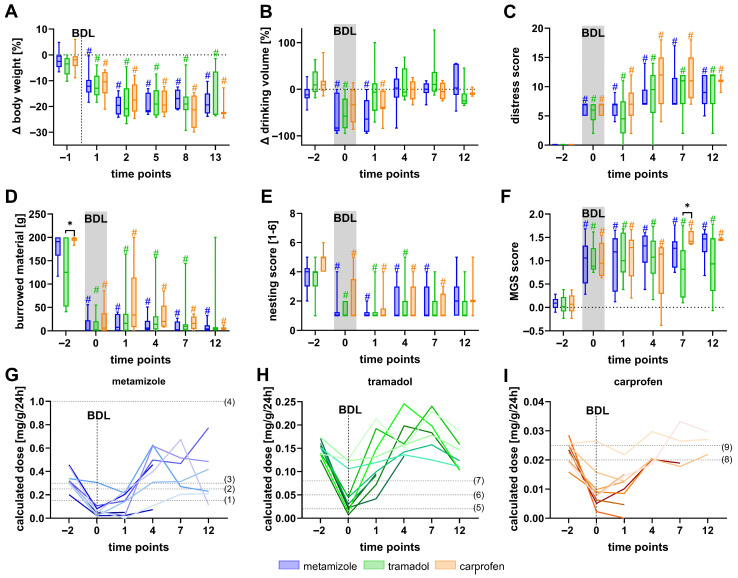
Behavioral and clinical indicators of distress following BDL under different analgesic treatments. BDL on day 0 resulted in reduced body weight (**A**) and drinking volume (**B**), increased distress score (**C**), decreased burrowing activity (**D**) and nesting behavior (**E**), and an elevated MGS score (**F**). Doses of metamizole (**G**), tramadol (**H**), and carprofen (**I**) were calculated and compared to the published effective doses (published doses = dashed lines, colors indicate individual mice). (1) Stumpf et al. [63], (2) Taylor et al. [65], (3) Schlosburg et al. [64], (4) Boros et al. [66], (5) Sacerdote et al. [67], (6) Mouedden et al. [69], (7) Wolfe et al. [70], (8) Matsumiya et al. [68] and (9) GV-Solas Committee [61]. Statistical analyses included mixed-effects analyses with Tukey’s correction for multiple comparisons between analgesic groups (* *p* ≤ 0.05), and Dunnett’s post hoc test for within-group comparisons relative to baseline (# *p* ≤ 0.05). n (metamizole) = 10, n (tramadol) = 10, n (carprofen) = 9. To see the number of animals contributing to the data at each time point, see Appendix A.

**Figure 7 biomedicines-13-03034-f007:**
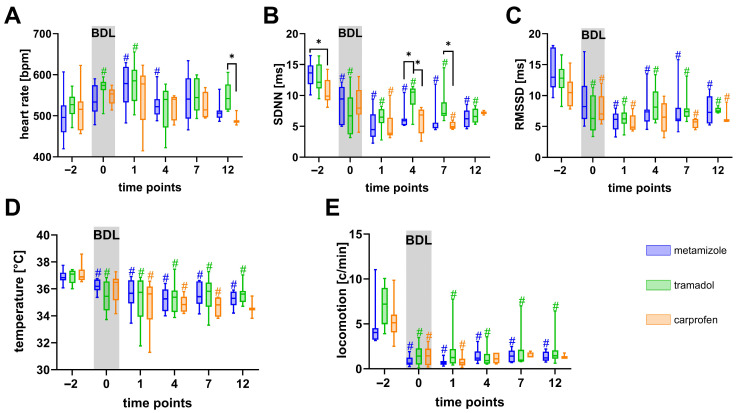
Physiological parameters after BDL recorded by telemetry. Following BDL, HR (**A**), SDNN (**B**), RMSSD (**C**), temperature (**D**), and locomotion (**E**) changed significantly. Statistical analyses included mixed-effects analyses with Tukey’s correction for multiple comparisons between analgesic groups (* *p* ≤ 0.05) and Dunnett’s post hoc test for within-group comparisons relative to baseline (# *p* ≤ 0.05). n(metamizole) = 10, n (tramadol) = 10, n (carprofen) = 9. To see the number of animals contributing to the data at each time point, see Appendix A.

**Figure 8 biomedicines-13-03034-f008:**
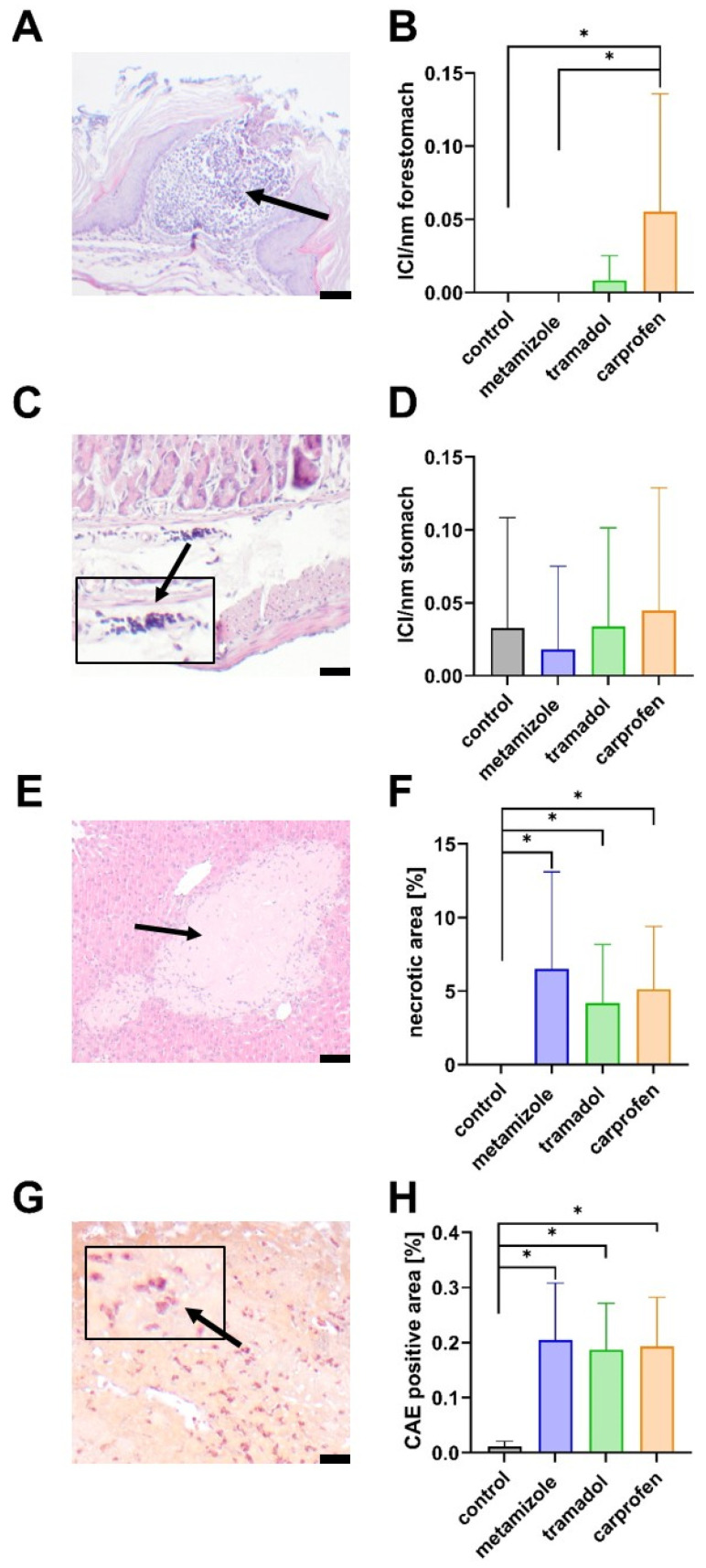
Histological evaluation of the stomach and liver after 14 days of cholestasis. Representative image of immune cell infiltrations (ICI, arrow) in the forestomach (**A**) and quantification across all groups (**B**). Immune cell infiltration (arrow) in the stomach (**C**) and quantification across all groups (**D**). Representative H&E-stained sections of a liver with pronounced necrosis (arrow) (**E**) and quantification of the extent of necrosis (**F**). Representative image of a CAE-stained (the arrow depicts red CAE^+^ cells) liver section (**G**) and the quantification of CAE+ area in the liver (**H**). The length of the scale bar is 50 µm (**A**,**E**) or 20 µm (**C**,**G**). Statistical analyses were performed using the Kruskal–Wallis test with Dunn’s correction for comparisons between analgesic groups (* *p* ≤ 0.05). n (control) = 10, n (metamizole) = 10, n (tramadol) = 9, n (carprofen) = 8 (**B**) or 9 (**D**).

## Data Availability

Rawdata_Analgesics.xlsx was provided during the review process and will be published after acceptance at Figshare: https://figshare.com/articles/dataset/Comparative_Evaluation_of_Analgesics_in_a_Murine_Bile_Duct_Ligation_Model/30555572?file=59355761 (accessed on 3 December 2025).

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
