# Peer review of "Comparative Evaluation of Analgesics in a Murine Bile Duct Ligation Model"

_biomedicines, 2025, doi:10.3390/biomedicines13123034_

Round 1

Reviewer 1 Report

Comments and Suggestions for Authors

The research paper entitled “Comparative Evaluation of Analgesics in a Murine Bile Duct Ligation Model”,  is a well-prepared paper and presents relevant data about the use of analgesics in preclinical disease models.

The study is methodologically sound, the data are reliable, and the results are consistent in the validated in vivo model. Statistical analysis is adequate, and results are reproducible and well represented.Discussion and conclusion sections are coherent, therefore, we only have a few minor revisions:

include method of euthanasia
Some figure panels could benefit from slightly larger axis labels and an increase in image quality or size.

Author Response

    1. include method of euthanasia

    We thank the reviewer for this important comment. We have now included the euthanasia method in the manuscript as follows: “Blood samples were collected via retro-orbital puncture under isoflurane anesthesia immediately before euthanasia by cervical dislocation.” (Line 229-230)

    In addition, we included the method as follows: “Additionally, euthanasia was performed under isoflurane anesthesia, if the combined distress score exceeded 15 points (with a theoretical maximum of 66 points).”  (Line 154-155)

    1. Some figure panels could benefit from slightly larger axis labels and an increase in image quality or size.

    Thank you for the valid comment. We increased the font of the axis labels and reduced the white space between the panels to increase the size of the figure panels. We have also improved the quality of all graphs to 1200 dpi.

Reviewer 2 Report

Comments and Suggestions for Authors

This manuscript presents a systematic comparison of three widely used analgesics, metamizole, tramadol, and carprofen, in the context of transmitter implantation and cholestatic injury induced by bile duct ligation (BDL) in mice. The study is ambitious in scope, integrating behavioral, clinical, physiological, and histopathological endpoints to evaluate animal welfare, and the authors are commended for the thorough multiparametric design and strong adherence to ethical refinement principles. The manuscript is well structured and clearly written. However, the following issues needs to be addressed before it is considered for publication.

  1. All mice, including those assigned to metamizole and tramadol, received perioperative carprofen injections at transmitter implantation and again before BDL. This introduces a substantial pharmacological confound because analgesic outcomes, behavioral changes, and even survival differences cannot be attributed solely to the orally administered agents, given that carprofen has a long half-life, biliary excretion, and cardiophysiological effects that may influence welfare endpoints. This substantially limits the interpretability of the comparative analgesic claims. The authors are encouraged to clearly emphasize this limitation in both Methods and Discussion, including expected pharmacokinetic overlap, and to temper or reframe any claims of drug superiority to reflect this unavoidable confounding factor.
  2. Postoperative reductions in water intake caused considerable inter-individual variability in actual mg/kg dosing, most prominently for metamizole and carprofen, resulting in many animals receiving sub-therapeutic doses throughout critical postoperative periods. This is a significant methodological concern because inadequate dosing, rather than intrinsic drug inefficacy, may underlie several reported welfare impairments, survival differences, or histological findings. The authors are encouraged to present individual mg/kg intake trajectories, quantify the proportion of animals below therapeutic thresholds per time-point, and explicitly acknowledge that dose variability may partially explain observed group differences.
  3. All welfare endpoints, distress score, burrowing, nesting, MGS, HR/HRV, temperature, locomotion, measure global well-being rather than pain specificity, and cannot discriminate nociceptive pain from cholestasis-induced malaise, sedation, stress, or sickness behavior. This matters because conclusions referring to “analgesic efficacy” may overstate what the metrics are able to demonstrate and risk conflating reduced distress with reduced pain. The authors are encouraged to bring more clarity on this and if required, revise the wording of their interpretations, emphasizing that the study assesses overall welfare rather than pain per se, and to introduce mechanistic caution regarding the non-specific nature of these metrics.
  4. While the Introduction provides comprehensive background on BDL, welfare assessment, and analgesics, it does not clearly articulate what is novel about the current comparative evaluation. This is important because high-impact journals require authors to clarify the scientific advance relative to existing literature and explain why this specific combination of telemetry, behavior, and histology provides new insight. The authors are encouraged to add a concise paragraph stating what gap in analgesic refinement this study fills, how it extends prior comparative studies, and why the BDL model uniquely requires evaluation of these three agents.
  5. Minor issues: Standardize “carprofen-treated” vs “carprofen treated.”
  6. Replace phrases like “problems with metamizole” with more neutral descriptions
  7. In discussion, claims of “tramodol superiority” should be softened; data show only modest advantages

Author Response

1. All mice, including those assigned to metamizole and tramadol, received perioperative carprofen injections at transmitter implantation and again before BDL. This introduces a substantial pharmacological confound because analgesic outcomes, behavioral changes, and even survival differences cannot be attributed solely to the orally administered agents, given that carprofen has a long half-life, biliary excretion, and cardiophysiological effects that may influence welfare endpoints. This substantially limits the interpretability of the comparative analgesic claims. The authors are encouraged to clearly emphasize this limitation in both Methods and Discussion, including expected pharmacokinetic overlap, and to temper or reframe any claims of drug superiority to reflect this unavoidable confounding factor.

We thank the reviewer for this important comment. We were obliged to perform additional perioperative analgesia to ensure optimal animal welfare during the perioperative phase. However, we agree that this subcutaneous carprofen administration represents a relevant pharmacological confound, as its relatively long half-life, biliary excretion, and cardiophysiological effects may influence behavioral, welfare-related, and survival outcomes.

In the methods section, we now included the following statements:

“All mice received one subcutaneous injection of carprofen (5 mg/kg; Rimadyl®, Pfizer GmbH, Berlin, Germany) before transmitter implantation and induction of cholestasis in addition to the analgesics administered via drinking water. While this approach likely enhanced overall pain management, it may also have confounded the assessment of the specific effects attributable to the orally supplied analgesics.” (Line 186 – 190)

In response to this comment, we have now explicitly addressed this issue also in the Discussion section, in which we describe the potential pharmacokinetic overlap and possible additive or synergistic interactions. (Line 644-658)

2. Postoperative reductions in water intake caused considerable inter-individual variability in actual mg/kg dosing, most prominently for metamizole and carprofen, resulting in many animals receiving sub-therapeutic doses throughout critical postoperative periods. This is a significant methodological concern because inadequate dosing, rather than intrinsic drug inefficacy, may underlie several reported welfare impairments, survival differences, or histological findings. The authors are encouraged to present individual mg/kg intake trajectories, quantify the proportion of animals below therapeutic thresholds per time-point, and explicitly acknowledge that dose variability may partially explain observed group differences.

We thank the reviewer for this valuable comment and fully acknowledge the importance of transparently presenting individual intake trajectories and therapeutic thresholds. As originally shown in Figures 3 and 6 (panels G–I), we had already visualized the calculated individual doses (mg/g/24 h) of ingested metamizole, tramadol, and carprofen.

To further improve clarity, we have now added Supplementary Tables S3–S5, which list the individual calculated doses (mg/kg/24 h) for each mouse at each time point. In addition, and in line with the reviewer’s recommendation, each table now includes a column indicating the number of animals that fell below the lowest defined therapeutic threshold at each time point. These tables are described in the Results section and referenced in the Discussion section to make this variability more explicitly visible. (Line 331-336, 406-413)

Furthermore, we have substantially expanded the section “Methodological considerations and limitations” to discuss more clearly how inconsistent water intake may lead to variable drug exposure and thereby complicate the interpretation of analgesic effects. We now state that apparent differences in welfare parameters may reflect differential dosing rather than true pharmacodynamic differences, and that both subtherapeutic underdosing and drug-related side effects must be considered when interpreting the findings. (Line 636-643)

3. All welfare endpoints, distress score, burrowing, nesting, MGS, HR/HRV, temperature, locomotion, measure global well-being rather than pain specificity, and cannot discriminate nociceptive pain from cholestasis-induced malaise, sedation, stress, or sickness behavior. This matters because conclusions referring to “analgesic efficacy” may overstate what the metrics can demonstrate and risk conflating reduced distress with reduced pain. The authors are encouraged to bring more clarity on this and if required, revise the wording of their interpretations, emphasizing that the study assesses overall welfare rather than pain per se, and to introduce mechanistic caution regarding the non-specific nature of these metrics.

We thank the reviewer for this comment and fully agree with this point. In response, we have avoided the use of terms such as “analgesic efficacy” when describing our data throughout the manuscript, except when describing other publications or in the context of the discussion, where we explicitly state that true analgesic efficacy cannot be determined without an untreated control group. This aspect is addressed in the section “Methodological considerations and limitations”. Furthermore, in Section 4.1, we clarify that all welfare indicators used in this study primarily reflect overall well-being rather than pain-specific states. We have also added text to explicitly emphasize that changes in these indicators observed after administration of analgesics are more appropriately interpreted as reductions in distress rather than as a specific reduction of pain.

See line 180, 499-501, 680-685

4. While the Introduction provides comprehensive background on BDL, welfare assessment, and analgesics, it does not clearly articulate what is novel about the current comparative evaluation. This is important because high-impact journals require authors to clarify the scientific advance relative to existing literature and explain why this specific combination of telemetry, behavior, and histology provides new insight. The authors are encouraged to add a concise paragraph stating what gap in analgesic refinement this study fills, how it extends prior comparative studies, and why the BDL model uniquely requires evaluation of these three agents.

We thank the reviewer for highlighting the need to clearly articulate the scientific novelty and refinement gap addressed by our study. In response, we have added a dedicated paragraph to the Introduction that explicitly outlines the lack of evidence-based comparative data on metamizole, tramadol, and carprofen in a cholestatic disease context. The revised text emphasizes that previous comparative studies have focused primarily on acute postoperative pain, whereas the BDL model involves both acute surgical pain and chronic cholestasis-associated distress, including inflammation and hepatocellular injury. We further clarify that chronic inflammation and chronic stress can alter pain processing and analgesic responsiveness. The newly added paragraph explains that this combination of postoperative and cholestatic pain represents a refinement gap and justifies the need for systematic comparison of the three widely used analgesics. Additionally, we describe how our integrated approach provides novel insight by capturing multiple dimensions of nociception and welfare under these complex conditions.

See Line 104-117

5. Minor issues: Standardize “carprofen-treated” vs “carprofen treated.”

Thanks for pointing that out. Now there is a hyphen everywhere.

6. Replace phrases like “problems with metamizole” with more neutral descriptions

Thank you for that comment. We agree and rewrote some sentences into more neutral descriptions. For example:

Old: “Overall, these results indicate that although tramadol has been the least problematic of the tested agents, it did not fully prevent reductions in animal welfare during cholestasis.”

New: “ Overall, these results indicate that although tramadol provided the most favorable welfare outcomes, it did not completely mitigate cholestasis-induced distress.” (Line 616-617)

7. In discussion, claims of “tramadol superiority” should be softened; data show only modest advantages

We agree with the reviewer’s point. In fact, we did not intend to describe tramadol as superior. We intended to emphasize that the data do not support a claim of superiority. To avoid any possible misunderstanding, we have revised the sentence to make this even clearer.

Old: “In addition, mice under tramadol had slightly lower MGS scores (Fig. 6F) and transiently higher SDNN (Fig. 7B), indicating minor benefits compared to the other analgesics. However, other assessment methods did not provide additional evidence that tramadol is the superior analgesic.”

Revised: “Although these results highlight modest, parameter-specific advantages of tramadol, the overall dataset does not provide strong evidence that it is substantially more beneficial than the other analgesics tested.” (Line 596-598)

Reviewer 3 Report

Comments and Suggestions for Authors

The manuscript presents a detailed and well-designed comparison of three analgesic regimens in the murine BDL model and provides high-value welfare-related insights for the 3Rs community. The experimental structure, combination of behavioral, clinical, physiological, and histopathological readouts is a clear strength.

However, a few points require clarification and minor improvements:

  1. Since all animals received a baseline carprofen injection before surgeries, please provide a brief explanation of how this may have influenced early postoperative analgesic comparisons.
  2. Drinking-water administration led to large inter-individual differences, especially for metamizole and carprofen. Consider strengthening the discussion on how inconsistent dosing may affect interpretation of analgesic efficacy.
  3. While literature ranges are cited, it would help to briefly explain why these exact concentrations (3 g/L, 1 g/L, 0.15 g/L) were chosen for this model.
  4. The manuscript states the sample size was calculated for body-weight change; please include the expected effect size and specific parameters used for statistical power.
  5. Because survival differed between groups, consider indicating the number of animals contributing data at each time point in the figure legends.
  6. The HRV differences between treatment groups (particularly lower values in carprofen-treated mice) should be further interpreted in light of known cardiovascular effects of COX inhibitors.
  7. The forestomach inflammation observed exclusively in the carprofen group is important; discuss potential mechanisms such as bile-dependent drug clearance impairment in cholestasis.
  8. The 22-day recovery period is substantial; briefly indicate whether welfare indicators fully normalized before BDL, or whether subtle residual effects may have remained.
  9. Increasing axis label font sizes or adding summary trend lines may help readers interpret multi-time-point data more easily.
  10. Since the study uses only male C57BL/6J mice, consider adding a short statement about how analgesic effects may differ in females or other strains.

Author Response

1. Since all animals received a baseline carprofen injection before surgeries, please provide a brief explanation of how this may have influenced early postoperative analgesic comparisons.

We thank the reviewer for this important comment. We agree that the perioperative administration of subcutaneous carprofen may influence postoperative analgesic comparisons.

In response to this comment, we have now explicitly addressed this issue in the manuscript. We added a dedicated paragraph in the “Methodological Considerations and Limitations” section of the Discussion, in which we describe the potential pharmacokinetic overlap and possible additive or synergistic interactions. (Line 644-658)

2. Drinking-water administration led to large inter-individual differences, especially for metamizole and carprofen. Consider strengthening the discussion on how inconsistent dosing may affect the interpretation of analgesic efficacy.

We thank the reviewer for this valuable comment and fully agree that inter-individual variability in drinking-water–based dosing is an important factor when interpreting analgesic effects. In response, we have strengthened the discussion. First, we have added Supplementary Tables S3–S5 in the results and discussion part, which now provide the individually calculated doses (mg/kg/24 h) for each mouse and each time point. These tables also indicate, for every day, how many animals fell below the lowest published therapeutic threshold. (Line 331-334, 406-413, 561-563)

Second, we have substantially expanded the section “Methodological considerations and limitations” to discuss more clearly how inconsistent water intake may lead to variable drug exposure and thereby complicate the interpretation of analgesic effects. We now state that apparent differences in welfare parameters may reflect differential dosing rather than true pharmacodynamic differences, and that both subtherapeutic underdosing and drug-related side effects must be considered when interpreting the findings. (Line 636-643)

3. While literature ranges are cited, it would help to briefly explain why these exact concentrations (3 g/L, 1 g/L, 0.15 g/L) were chosen for this model.

Thank you for this helpful comment. The selected doses are based on the Expert Information from the GV-SOLAS Committee for Anaesthesia, developed in collaboration with Working Group 4 of the TVT (“Pain management for laboratory animals”). In addition, in our previous studies across different disease models, we have consistently used 3 g/L metamizole in the drinking water rather than 1.25 g/L (e.g., PMID: 38791980; doi: 10.3390/ani10122306). We have now added these references to the manuscript in the materials and methods sections under 2.4 to justify the chosen concentrations. (Line 183 – 186)

4. The manuscript states the sample size was calculated for body-weight change; please include the expected effect size and specific parameters used for statistical power.

We now included expected effect size and specific parameters in the methods section. See lines 134-136

5. Because survival differed between groups, consider indicating the number of animals contributing data at each time point in the figure legends.

We thank the reviewer for this helpful suggestion. We agree that reporting the number of animals contributing data at each time point is important for interpreting the results. To maintain clarity and avoid overly long figure legends or more complicated figures, particularly given the three analgesic groups and multiple time points, we have created Supplementary Table 6, which provides the exact number of animals remaining in each group at each time point. This approach ensures full transparency while preserving the readability of the main figures. We refer in the legends of Fig. 6 and Fig. 7 to supplementary Table 6. (See Line 424-425, 444-445)

6. The HRV differences between treatment groups (particularly lower values in carprofen-treated mice) should be further interpreted in light of known cardiovascular effects of COX inhibitors.

We thank the reviewer for this helpful comment. We have expanded the discussion in 4.3 to more comprehensively address the potential cardiovascular mechanisms underlying the reduced HRV observed in carprofen-treated mice. Specifically, we now emphasize the central role of COX-derived prostaglandins (PGEâ‚‚, PGIâ‚‚) in vascular tone, renal perfusion, and blood pressure homeostasis, and discuss how COX-2 inhibition can elevate arterial pressure. These hemodynamic changes may shift autonomic balance toward sympathetic dominance, thereby decreasing HRV. (Line 544-559)

7. The forestomach inflammation observed exclusively in the carprofen group is important; discuss potential mechanisms such as bile-dependent drug clearance impairment in cholestasis.

We agree that the inflammation in the carprofen-treated group is important. We have expanded the discussion to address potential mechanisms underlying the forestomach inflammation observed in the carprofen group. Specifically, we now highlight that carprofen is predominantly cleared via biliary excretion, and that bile duct ligation markedly reduces bile flow, thereby impairing drug clearance and likely increasing systemic and local exposure to carprofen. We further discuss how prolonged exposure and alterations in enterohepatic circulation due to cholestasis may enhance toxicity. These additions are included in the manuscript under 4.4.  (Line 606-615).

8. The 22-day recovery period is substantial; briefly indicate whether welfare indicators fully normalized before BDL, or whether subtle residual effects may have remained.

We thank the reviewer for this important comment. The 22-day recovery period was chosen based on Kumstel et al. (https://doi.org/10.1016/j.jare.2019.09.002), who demonstrated that physiological and behavioral parameters typically stabilize 7–13 days after transmitter implantation. Therefore, we used three weeks to ensure that all parameters would stabilize.

In our study, distress score, nesting, and burrowing had returned to baseline before Transmitter implantation. However, two deviations remained detectable in the tramadol group: body weight was still slightly reduced compared with the pre-operative phase, whereas water intake was unexpectedly increased. We now acknowledge that subtle residual effects may have remained. (Line 537-543)

9. Increasing axis label font sizes or adding summary trend lines may help readers interpret multi-time-point data more easily.

Thank you for the valid comment. We increased the font of the axis labels, reduced the white space between the panels to increase the size of the figure panels. We have also improved the quality of all graphs to 1200 dpi.

10. Since the study uses only male C57BL/6J mice, consider adding a short statement about how analgesic effects may differ in females or other strains.

Thank you for this valuable suggestion. We have added a concise statement to the Discussion in 4.5, noting that analgesic responses can vary substantially across mouse strains and between sexes. We now explicitly acknowledge that the use of only male C57BL/6J mice limits the generalizability of our findings to other strains or to female animals. (Line 659-669)

Round 2

Reviewer 2 Report

Comments and Suggestions for Authors

All comments has been addressed to the satisfaction

Author Response

thank you for your review. Based on the comment of the editor we changed in line 138 Helicobacter pylorii to helicobater spp.